# Prognostic role of the prognostic nutritional index (PNI) in patients with head and neck neoplasms undergoing radiotherapy: A meta-analysis

**Yujie Shi, Yue Zhang, Yaling Niu, Yingjie Chen, Changgui Kou** [ORCID] *

Department of Epidemiology and Biostatistics, School of Public Health, Jilin University, Changchun, Jilin Province, China

* koucg@jlu.edu.cn

## Abstract

### Background

This novel meta-analysis was conducted to systematically and comprehensively evaluate the prognostic role of the pretreatment PNI in patients with head and neck neoplasms (HNNs) undergoing radiotherapy.

### Methods

Three databases, PubMed, Embase, and Web of Science, were used to retrieve desired literature. Hazard ratios (HRs) with 95% confidence intervals (CIs) were extracted and pooled by fixed-effects or random-effects models to analyze the relationship between the PNI and survival outcomes: overall survival (OS), distant metastasis-free survival (DMFS), and progression-free survival (PFS).

### Results

Ten eligible studies involving 3,458 HNN patients were included in our analysis. The robustness of the pooled results was ensured by heterogeneity tests ($I^2$ = 22.6%, 0.0%, and 0.0% for OS, DMFS, and PFS, respectively). The fixed-effects model revealed a lower pretreatment PNI was significantly related to a worse OS (HR = 1.974; 95% CI: 1.642–2.373; P<0.001), DMFS (HR = 1.959; 95% CI: 1.599–2.401; P<0.001), and PFS (HR = 1.498; 95% CI: 1.219–1.842; P<0.001). The trim-and-fill method (HR = 1.877; 95% CI: 1.361–2.589) was also used to prove that the existing publication bias did not deteriorate the reliability of the relationship.

### Conclusion

The pretreatment PNI is a promising indicator to evaluate and predict the long-term prognostic survival outcomes in HNN patients undergoing radiotherapy.

**Data Availability Statement:** All relevant data are within the manuscript and its Supporting Information files.

**Funding:** The authors received no specific funding for this work.

**Competing interests:** The authors have declared that no competing interests exist.

## Introduction

Head and neck neoplasms (HNNs), a spectrum of diseases including neck tumors, otolaryngology tumors, and oral and maxillofacial tumors, always show a noticeably different response to standard treatments and exhibit diversified prognoses [1]. A relatively worse and uncertain 5-year survival rate for HNN patients, varying from 30% to 70% based on the stage and location of the tumor, has been reported by Hoesseini et al. [2]. The standard therapies for initial and locally advanced malignant HNNs have long been considered radiotherapy and surgery with or without chemotherapy [3]. Although the recent two decades have seen the rapid development of radiotherapy technology, some inevitably severe side effects profoundly impact patient survival outcomes after treatment, such as oral mucositis, dysgeusia, and dysphagia [4]. Given the potential hazards of radiotherapy, discussing prognoses before treatment could allow a well-considered adaptation of remedies and ultimately benefit HNN patients.

Presently, the tumor-node-metastasis (TNM) staging system, histological subtype, and genetic biomarkers are widely regarded as common tools to help physicians estimate the pretreatment conditions of patients, particularly TNM classification. However, the fact that some patients diagnosed with the same tumor type and identical TNM stage still facing diverse prognoses in clinic has been always warning us. Hence, it is extremely necessary to identify another effective indicator to help predict survival outcomes before treatment in HNN patients [5, 6]. The prognostic nutritional index (PNI), calculated with the serum albumin levels and total lymphocyte count in peripheral blood [7], has been firmly identified as an excellent indicator for assessing and predicting survival outcomes in patients with different cancers, such as lung cancer [8] and gastric cancer [9]. However, the relationship between the pretreatment PNI and the prognostic situation of HNN patients undergoing radiotherapy has still remained confusing and ambiguous, without a unified and integrated conclusion yet. Therefore, this meta-analysis aimed to resolve this issue.

## Materials and methods

### Search strategy and study selection

This meta-analysis was performed under the guidelines of the Preferred Reporting Items for Systematic Reviews and Meta-Analyses (PRISMA) and was registered in the International Prospective Register of Systematic Reviews (PROSPERO) (registration number: CRD42020223643). We comprehensively searched three main databases, PubMed, Web of Science, and Embase, using a combination of MeSH and entry terms. The exact electronic search strategy has been attached in (S1 File). These search strategies were determined using multiple preretrieval tests and would be slightly adjusted when used in different databases. We also conducted expanded search at the same time to ensure that we can retain all of the relevant articles involving all kinds of sub types of head and neck neoplasms. All qualified English publications from inception to November 30th, 2020 were searched for. The references of all eligible articles were also independently screened to identify additional studies excluded in the initial search.

### Inclusion and exclusion criteria

The inclusion criteria were as follows: (1) patients were diagnosed with HNNs histopathologically; (2) patients were treated with radiotherapy; (3) the association between pretreatment PNI and OS, PFS, and/or DMFS was recorded.

Following articles types, including case reports, letters, reviews, conference abstracts, and articles published in only abstract form, as well as articles written in non-English were all

crossed off. Articles with HRs for survival outcomes not provided directly were excluded as well.

## Quality assessment

The Newcastle-Ottawa Scale (NOS) was used to assess the methodological quality of all the eligible studies [10]. In the NOS system, 9 is the most perfect score including 4 out of 9 for Selection, 2 for Comparability, and 3 for Outcomes. Studies with a score greater than or equal to 6 are generally considered high quality [6]. The process was conducted by 2 researchers. If any discrepancy appeared, they would discuss with each other, and went back to review the article again. If it didn't work by discussion to reach an agreement, another researcher would be invited to evaluate the quality of the article independently and finally make a decision.

## Data extraction

Two researchers independently participated in extracting data by utilizing a well-designed data collection Excel sheet prepared in advance. If any discrepancies appeared, a third person was invited to extract the data again and then discussed with the above two colleagues for a consensus. The data extracted obtained items such as the followings: study; accrual period; country; study design; tumor; number (male/female); age (median); treatment; PNI cutoff value; follow-up (month); outcome; cutoff value determination; HRs and 95% CIs for OS, DMFS and/or PFS.

## Statistical analysis

All the data were processed using STATA 15.1 software. We adopted pooled HRs and their 95% CIs calculated by a fixed-effects model or random-effects model as the final effect size. Heterogeneity was evaluated using the $I^2$ statistic. An $I^2$ less than 50% indicated no significant heterogeneity in place; thus, a fixed-effects model was available for pooling HRs and 95% CIs. Conversely, if statistical heterogeneity existed ($I^2 > 50\%$), sensitivity analysis and stratified analysis were required to determine the sources of heterogeneity, and we preferred to select a random-effects model to combine HRs. Additionally, publication bias was assessed by funnel plots and Egger's test; $P > 0.05$ indicated no significant bias in publication. Otherwise, the trim-and-fill method was used to evaluate to what extent this bias affected the results.

# Results

## Retrieval of literature and study characteristics

Fig 1 shows the entire literature search process. We first sought 242 possibly relevant articles in three major databases: PubMed, Web of Science, and Embase.

After deleting duplicate articles and looking through titles and abstracts, we retained 32 studies. Ultimately, based on careful full-text reviews, only 10 cohort studies published from 2015 to 2020 [11–20] were regarded as valuable for this meta-analysis (Table 1).

Of the 10 included studies, 7 were conducted in China [12–14, 16–19], and the remaining 3 were conducted in Japan [20], Turkey [15] and Spain [11], respectively. Among ten studies, six were performed in patients with nasopharyngeal carcinoma (NPC) [13, 15–19], and one each in patients with oral squamous cell carcinoma [20], locoregionally advanced squamous cell head and neck cancer [11], cervical esophageal squamous cell carcinoma [12], and laryngeal cancer [14]. All the patients were treated with radiotherapy such as CRT and IMRT. The cutoff value of PNI, ranging from 42.7 to 55, was determined by 3 methods: ROC curve, Cutoff

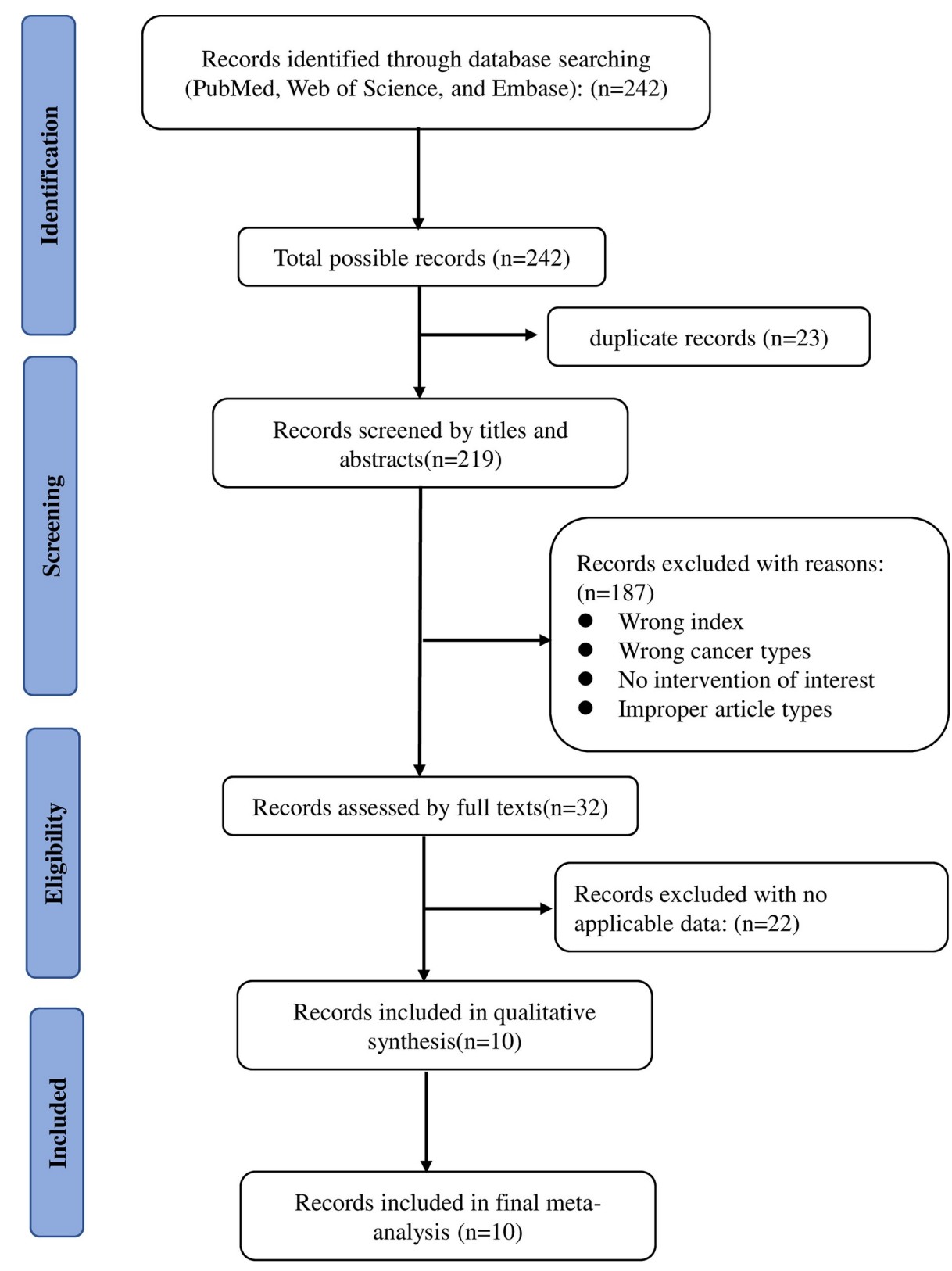

**Fig 1. Flow diagram of study selection procedure.**

**Table 1. The baseline characteristics of the included studies.**

| Study | Accrual period | Country | Study design | Tumor type | Number (male/ female) | Age (Median) | Treatment | PNI cut-off value | Follow-up (month) | Outcomes | Cut-off value determination | QS |
|---|---|---|---|---|---|---|---|---|---|---|---|---|
| Mete 2019 | 2010–2018 | Turkey | cohort | NPC[a] | 95 (67/28) | 50 | IMRT[f] | 45.45 | 41.0 | OS[l] DMFS[m] | ROC curve | 9 |
| Miao 2017 | 2001.4–2010.06 | China | cohort | NPC | 270 | NR[o] | IMRT | 52 | 109.50 | OS DMFS | ROC curve | 7 |
| Du 2015 | 2003.01–2006.12 | China | cohort | NPC | 694 (517/177) | 44 | RT[g] | 55 | 88 | OS DMFS PFS[n] | Median | 8 |
| Lin 2016 | 2007.10–2009.12 | China | cohort | NPC | 1168 (853/315) | 47 | NEO[h]+RT NEO+CCRT[i] | 51 | 68.8 | DMFS | Cut-off Finder | 8 |
| He 2019 | 2010.12–2017.6 | China | cohort | NPC | 337 (271/106) | 47 | NEO+CCRT | 49.05 | 40 | OS PFS | Cut-off Finder | 8 |
| Ronald 2018 | 2010.1–2013.12 | China | cohort | NPC | 585 (420/165) | 49 | IMRT | 53 | 63.3 | OS PFS DMFS | ROC curve | 9 |
| Gema 2018 | 2010.5–2016.5 | Spain | cohort | LAHNSCC[b] | 95(90/5) | 60(mean) | ICT[j]+CCRT | 45 | 29.1 | OS | ROC curve | 7 |
| Dai 2019 | 2000.6–2015.12 | China | cohort | CESCC[c] | 106(79/27) | 58 | CRT[k] | 48.15 | 19.5 | OS | Cut-off Finder | 7 |
| Fu 2019 | 2009.1–2014.6 | China | cohort | LC[d] | 61(59/2) | 57.2 | RT | 44 | NR | OS | ROC curve | 7 |
| Ryoji 2020 | 2004.01–2011.12 | Japan | cohort | OSCC[e] | 47(23/24) | 79 | CRT | 42.7 | NR | OS PFS | ROC curve | 8 |

[a]**NPC:** nasopharyngeal carcinoma

[b]**LAHNSCC:** locoregionally advanced squamous cell head and neck cancer

[c]**CESCC:** cervical esophageal squamous cell carcinoma

[d]**LC:** laryngeal cancer

[e]**OSCC:** oral squamous cell carcinoma

[f]**IMRT:** intensity modulated radiotherapy

[g]**RT:** radiotherapy

[h]**NEO:** neoadjuvant chemotherapy

[i]**CCRT:** concurrent chemoradiation

[j]**ICT:** induction chemotherapy

[k]**CRT:** chemoradiotherapy

[l]**OS:** overall survival

[m]**DMFS:** distant metastasis-free survival

[n]**PFS:** progression-free survival

[o]**NR:** not report.

Finder (a web application), and median. All the eligible studies were assessed with a score $\geq 6$ according to the NOS system, confirming they were all of high quality.

## Pooling analysis

**Association between PNI and OS.** Among all the 10 studies, nine evaluated the relationship between the pretreatment PNI and OS [11–18, 20], and provided valid HRs and 95% CIs. The $I^2$ statistic ($I^2 = 22.6\%$) indicated that no significant heterogeneity existed among these studies. Thus, combined with the fixed-effects model, HNN patients with lower pretreatment PNI undergoing radiotherapy, as shown in Fig 2A, were more likely to face a higher risk of death in the long term (OS: HR = 1.974; 95% CI: 1.642–2.373; P<0.001).

**Association between PNI and DMFS.** Five of ten eligible articles reported the value of DMFS [13, 15, 17–19] and presented the HRs and 95% CIs. Because no significant

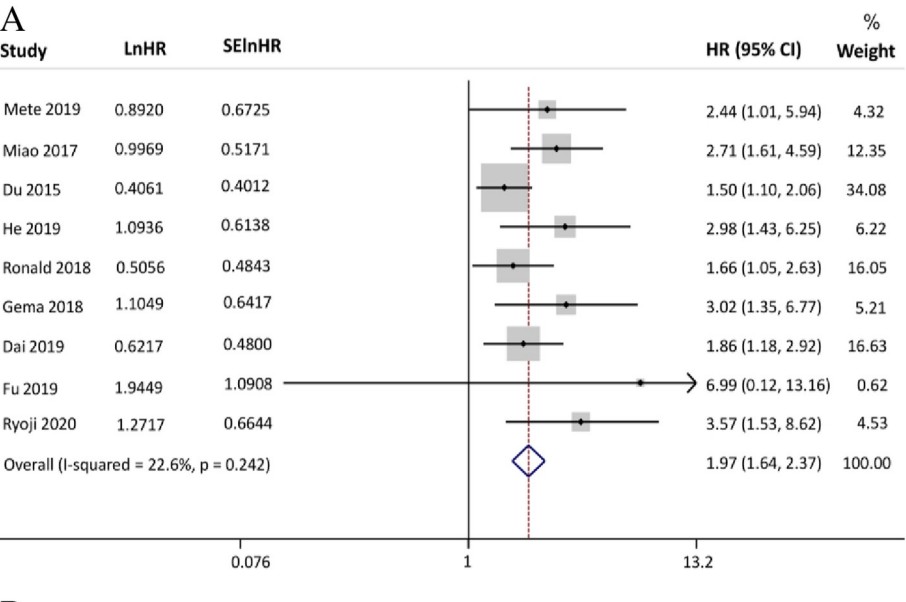

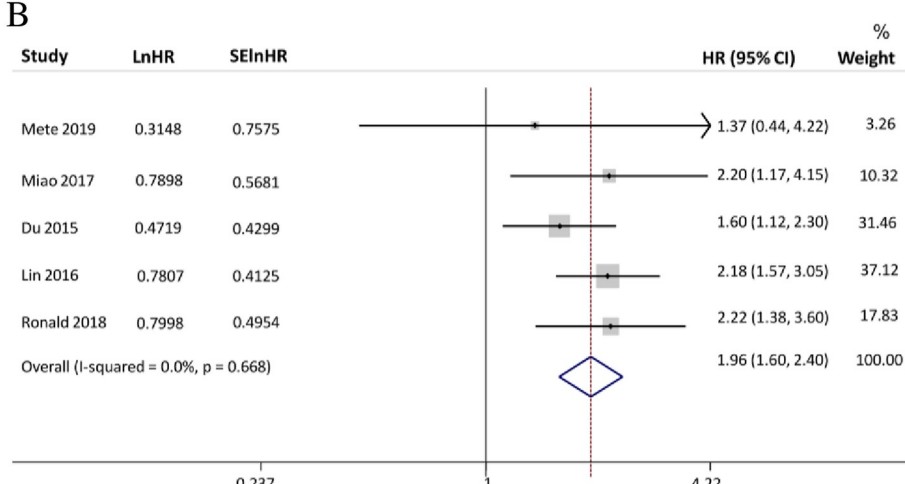

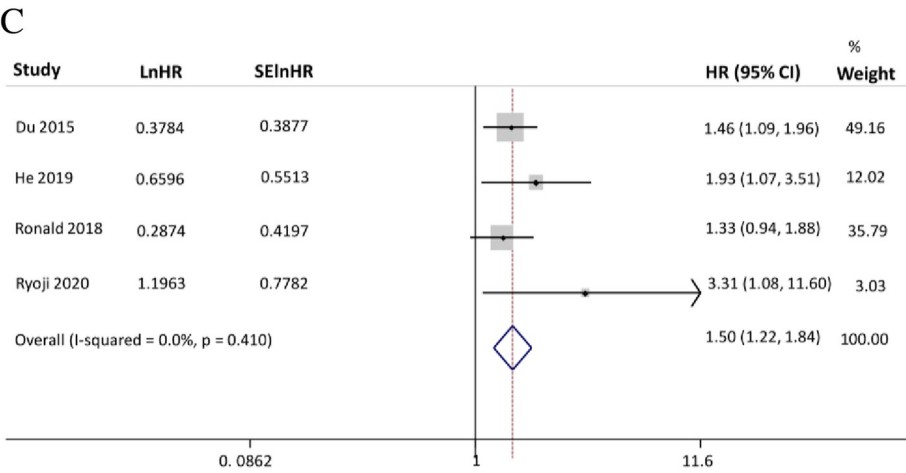

**Fig 2.** Forest plots of the relationship between the PNI and survival outcomes: A. OS B. DMFS C. PFS.

heterogeneity was detected ($I^2 = 0.0\%$), a fixed-effects model was available, and the results illustrated that decreased pretreatment PNI was closely associated with worse DMFS (HR = 1.959; 95% CI: 1.599–2.401; P<0.001) (Fig 2B).

**Association between PNI and PFS.** The indicator of PFS was recorded in four articles [13, 16, 18, 20], and the HRs and 95% CIs were also extracted. Testing result of $I^2$ statistic ($I^2 = 0.0\%$) clearly manifested homogeneity among these studies. Therefore, a fixed-effects model was still adopted to synthesize HRs. As shown in Fig 2C, an inferior PNI strongly suggested the occurrence of early cancer progression (PFS: HR = 1.498; 95% CI: 1.219–1.842; P<0.001).

**Sensitivity analysis and publication analysis.** To evaluate the robustness of the pooled HRs for OS, DMFS and PFS, we performed sensitivity analysis by pooling HRs after omitting one study at a time. As shown in Fig 3, no obvious changes were observed in the pooled HRs before and after deleting any single study for OS, DMFS and PFS, suggesting the robustness and reliability of the results. Because of the limited number of eligible studies for DMFS (5 articles) and PFS (4 articles), we did not detect publication bias for them. As for the indicator of OS (9 articles), we conducted Egger's test to help analyze potential existing bias on publication. The funnel plot (Fig 4A) showed an asymmetric distribution, and Egger's quantitative test displayed P<0.05, both implying publication bias. To assess to what extent publication bias affected the pooled result, we performed the trim-and-fill method (Fig 4B). The new funnel plot presented a symmetric distribution, and the pooled HR after trimming and filling was 1.877 (95% CI: 1.361–2.589), in line with the initial result (HR = 1.974; 95% CI: 1.642–2.373). Thus, our pooled HR initially using the fixed-effects model was sufficiently robust and credible without interference by such unimpressive publication bias.

## Discussion

Because of the existing disadvantages of current grade standards for cancer patients, such as TNM classification, it is urgently necessary to explore a new, more effective, and individualized indicator to help physicians predict the possible prognoses after treatment. In the past two decades, an increasing attention has been given to the prognostic nutritional index (PNI) to investigate its relationship with different cancers [21–23]. To date, prognostic outcomes of many types of neoplasms, such as gastric carcinoma and lung cancer, have been identified being closely related to PNI value [8, 9], while the association between the prognoses in HNN patients undergoing radiotherapy and the pretreatment PNI has not been reported integratedly and systematically yet. To our best knowledge, this meta-analysis is the first to resolve this problem comprehensively.

In our meta-analysis, 10 high-quality studies (NOS score ≥ 6) involving 3,458 HNN patients treated with radiotherapy were included. A significant relationship was found between the PNI and OS, DMFS, and PFS, suggesting an HNN patient with a lower pretreatment PNI would be more inclined to possess a worse prognosis after radiotherapy. $I^2$ statistic revealed the heterogeneity in these studies were insignificant, indicating sufficient credibility of the pooled results. Based on Fig 3 of the sensitivity analysis, we could confirm the robustness of results. Meanwhile, we also noticed the article (Du et al. [13]) had the biggest impact on final HRs. After reviewing and comparing these studies, we found that the research of Du et al. was the only one that determined the cutoff value of PNI using the median rather than a ROC curve or the Cutoff Finder. In his study, the cutoff value of PNI, 55, remained the highest comparing with the other 9 researches, and the median age of research patients, 44 years, was generally lower than that in others. These factors likely caused the prolongation of OS, DMFS, and PFS in the lower PNI group and then shortened the gap between the two contrasting groups, so the HR we got in this study would be slightly closer to 1 than others.

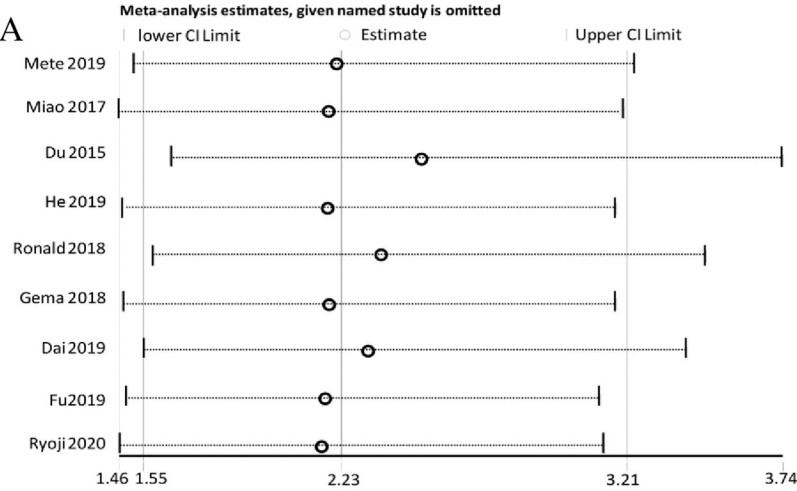

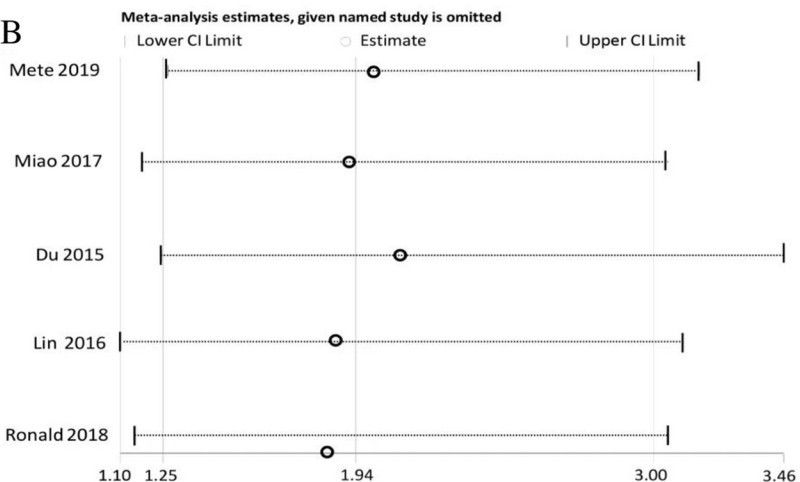

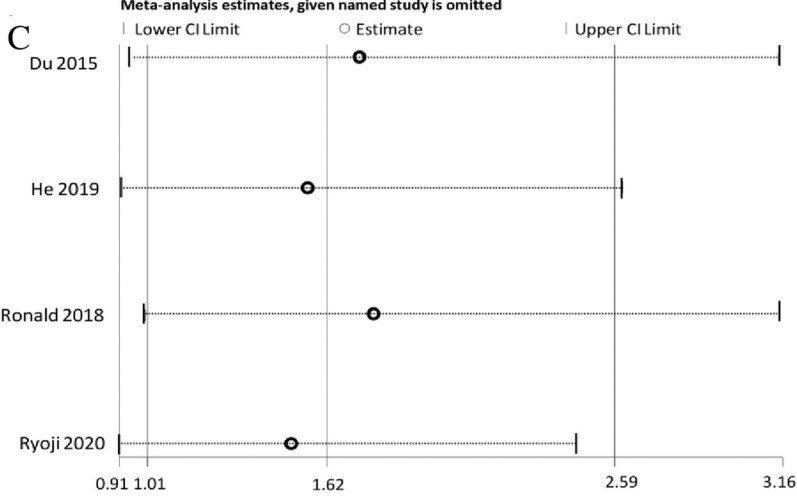

**Fig 3.** Sensitivity analyses of HRs: A. OS B. DMFS C. PFS.

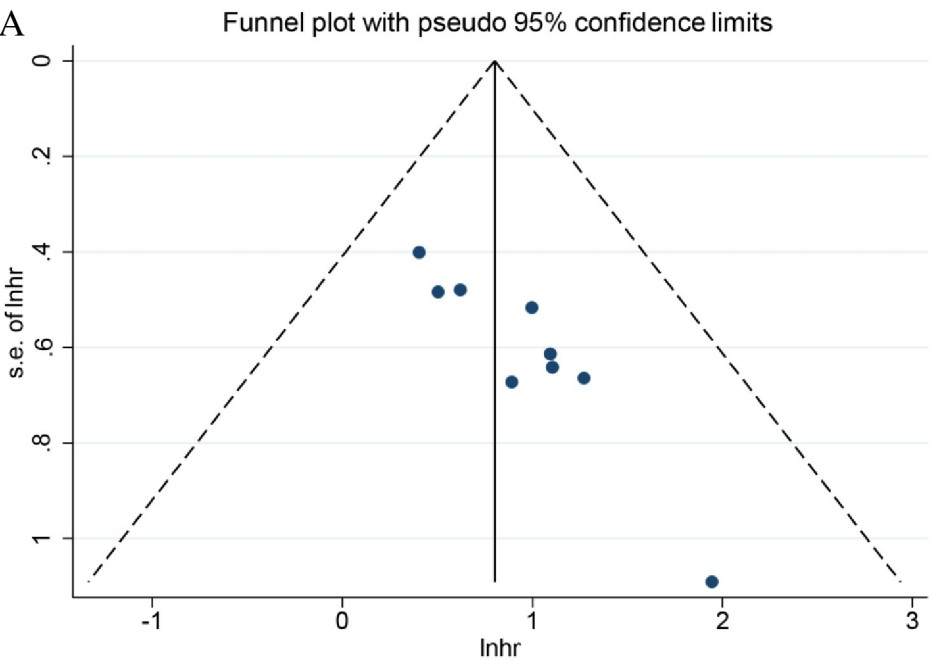

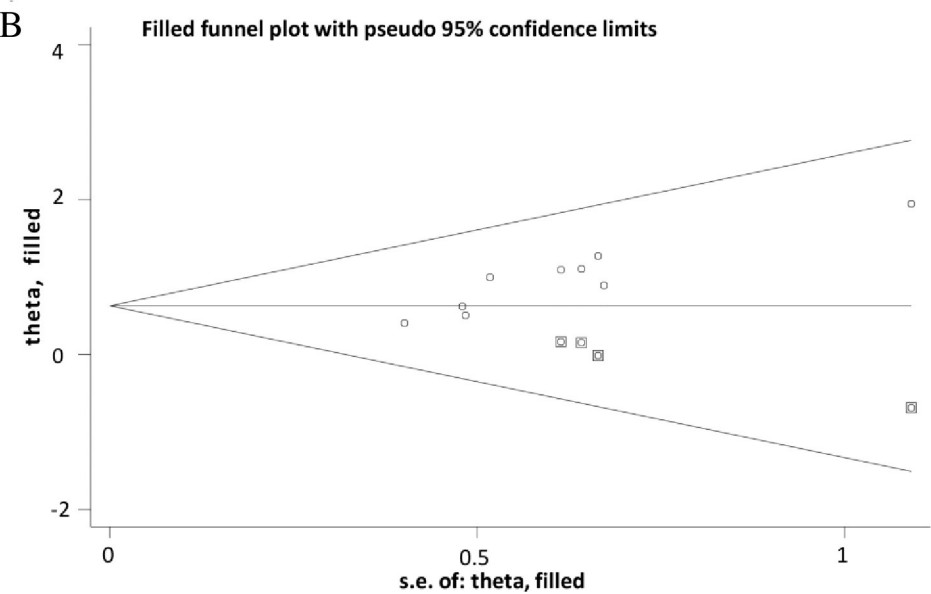

**Fig 4.** Test results of publication bias: A. before trim-and-fill method B. after trim-and-fill method.

For publication bias, although Egger's test indicated significant publication bias in place, the funnel plot adjusted by the trim-and-fill method still showed symmetry (Fig 4), and the pooled HR and 95% CI kept pace with the initial ones obtained by the fixed-effects model (both greater than 1), indicating that this publication bias did not undermine the reliability of our pooled results. After discussion, we thought the limited included literature and unequal publication status of articles with positive and negative results led to this bias in our study. All the results together indicated that the pretreatment PNI is indeed a promising indicator to predict prognostic survival outcomes of HNN patients undergoing radiotherapy.

The PNI is calculated with serum albumin and lymphocytes in peripheral blood [PNI = 10*albumin value (g/dl) + 0.005*total lymphocyte count in peripheral blood (per mm$^3$)] [24], making it not difficult to explain why PNI could reflect prognoses. First, serum albumin and lymphocytes in peripheral blood represent the body's nutritional and immunological status, respectively. The study of Tang et al. [25] reported that malnutrition is very common in cancer patients, with a prevalence varying from 39% to 71%, and it is always accompanied by a higher incidence of infections, treatment-related toxicities, and a consequent longer hospital stay. As we know, our immune system represents the body's ability to monitor and clear both outside invaders (such as external viruses) and inside mutant cells (such as cancer cells) [26, 27]. A poor nutritional or immune status often results in an obvious decline in the body's defense capability, leading to frequent local tumor recurrence and/or distant metastasis, and finally shortens the time of progression-free survival (PFS), distant metastasis-free survival (DMFS), and ultimate overall survival (OS) of patients [28]. Second, at present, increasing evidence has elucidated that patients with a low nutritional status often fail to tolerate the whole course of radiotherapy or chemotherapy treatment [29]. The delay or suspension of remedies could also lead to disease deterioration. Tang et al. [6] reported that radio-sensitivity and chemical sensitivity show a noticeable decline in patients with lower PNI, compromising the therapeutic effect to different degrees and subsequently reducing the final survival time. Third, the solid relationship between the lower PNI and advanced tumor clinical features, including older age, advanced TNM stage, deeper depth of cancer, and others that determine poor prognoses, has been increasingly confirmed in many studies [16, 17]. In addition to above reasons, PNI was also characterized by its convenience and easy acquisition. Thus, every cancer patient could have access to a PNI evaluation before receiving radiotherapy and then treatment plans could be adjusted individually. Given the above, the PNI is indeed a reasonable and practical prognostic indicator. However, despite such reasons in place, we still cannot conclude that the low PNI explains the poor prognostic outcomes in HNN patients receiving radiotherapy. Further prospective and profound studies are expected to provide more tenable supports. Meanwhile, PNI threshold is really essential when we promote this indicator in clinic to make a prognostication. It is safer for a patient whose PNI is greater than the threshold to receive radiotherapy in statistics. It is a cut-off point currently calculated by three main approaches: ROC curve, Cutoff Finder (a web application), and PNI median. Among them, ROC curve has been gradually recognized as one of the most promising methods, and has been used widely in practice. However, a clear and unified cutoff value of PNI has yet to be decided so far, and it varies in a wide range in different studies (42.7 to 55 in this meta-analysis). That would be a great constraint when we promote this indicator in clinic. Hence, more evidence is urgently required to figure out what is the optimal cutoff value of the PNI.

This study had potential limitations that must be addressed. First, all 10 eligible cohort studies were conducted retrospectively, indicating selection bias, withdraw bias and others inevitably existed. Thus, it would be better to seek additional prospective studies to support our results in the future. Second, our results might be slightly inaccurate when applied in non-Asian countries. Nine of the 10 included studies were performed in Asia (China, Japan, and Turkey). Thus, more researches in American and European countries concerning prognostic role of the PNI in HNN patients are expected. Third, the number of included studies for DMFS (5 articles) and PFS (4 articles) was limited, potentially overestimating or underestimating the true effect of the predictive function of the PNI. Finally, according to existing surveys, except for thyroid tumors, other head and neck neoplasms always occur more frequently in male patients [30]. Additionally, age is usually associated with a person's nutritional and immune condition [21]. Thus, sex and age are potential confounding factors for the results.

However, because of the limited data we collected, we are incapable of conducting subgroup analysis based on sex and age. Therefore, whether the prognostic role of the PNI varies based on such factors must be explored further.

## Conclusion

The PNI is a promising indicator to predict prognostic survival outcomes of HNN patients undergoing radiotherapy. HNN patients with a lower pretreatment PNI are more likely to face a significantly worse OS, DMFS, and PFS, which offers an opportunity to improve the patients' nutritional and immune status before radiotherapy. Future researches are needed to help identify an optimal cutoff value of the PNI, which will guide clinicians to make more precise and individualized therapeutic plans and further improve patients' survival outcomes.

## Supporting information

**S1 File. Electronic search strategy.**
(PDF)

**S2 File. Data availability.**
(XLSX)

## Acknowledgments

We sincerely thank Wei Bai and Shoumeng Yan researchers for their help provided during this process.

## Author Contributions

**Conceptualization:** Yujie Shi, Changgui Kou.

**Data curation:** Yujie Shi.

**Formal analysis:** Yujie Shi, Yue Zhang, Yaling Niu, Yingjie Chen.

**Methodology:** Yujie Shi, Changgui Kou.

**Software:** Yujie Shi.

**Supervision:** Changgui Kou.

**Writing – original draft:** Yujie Shi, Yue Zhang, Yaling Niu.

**Writing – review & editing:** Yujie Shi, Changgui Kou.

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
