## [Decision Letter · Decision Letter 0]

7 Jul 2021

PONE-D-21-13461

Prognostic role of the prognostic nutritional index (PNI) in patients with head and neck neoplasms undergoing radiotherapy: A meta-analysis

PLOS ONE

Dear Dr. Kou,

Thank you for submitting your manuscript to PLOS ONE. After careful consideration, we feel that it has merit but does not fully meet PLOS ONE’s publication criteria as it currently stands. Therefore, we invite you to submit a revised version of the manuscript that addresses the points raised during the review process.

We look forward to receiving your revised manuscript.

Kind regards,

Mona Pathak, PhD

Academic Editor

PLOS ONE

Additional Editor Comments:

The objective of the manuscript is very much relevant. I have biggest concern that search strategy was not appropriate to retrieve all relevant studies. Further, exact electronic search strategy should also be reported. Authors need to review their electronic search strategy and make it more sensitive. Authors should refer to latest version of PRISMA guideline released in 2020. Please address the concerns raised by reviewers.

Journal Requirements:

Reviewers' comments:

Reviewer's Responses to Questions

**Comments to the Author**

1. Is the manuscript technically sound, and do the data support the conclusions?

Reviewer #1: Yes

Reviewer #2: Partly

Reviewer #3: Yes

2. Has the statistical analysis been performed appropriately and rigorously? 

Reviewer #1: Yes

Reviewer #2: Yes

Reviewer #3: Yes

3. Have the authors made all data underlying the findings in their manuscript fully available?

Reviewer #1: Yes

Reviewer #2: Yes

Reviewer #3: No

4. Is the manuscript presented in an intelligible fashion and written in standard English?

Reviewer #1: Yes

Reviewer #2: Yes

Reviewer #3: Yes

5. Review Comments to the Author

Reviewer #1: Dear Authors,

Congratulations with your article. It is an nice overview about the role of the PNI in patients with head and neck neoplasms undergoing radiotherapy. However some improvements are mandatory for publication:

1. Please comment on the PNI threshold, should and how should it clinically be used?

2. Your conclusion should be rewritten and be formulation more strongly, you’ve looked into all the articles, so you might have an profound opinion about the PNI.

3. Small comment: please remove the first n=0 in the block diagram and please mention the three databases used. Furthermore this is a clear and good diagram.

Reviewer #2: I am little worried about the search strategy. Head and Neck are broad spectrum of neoplasms with differing morphologies, which includes mainly upper aerodigestive tract (oral cavity, paranasal sinuses, pharynx, larynx, and cervical esophagus), thyroid, associated lymph nodes, soft tissues, and bone etc. Even, oral cavity includes many sub types like lip, tongue, the roof and floor of the mouth etc. Therefore, there is possibility that, articles may included word head and neck or may not and only use name of specific sub type. How these were managed in the search process is not clear and not clearly indicated in the manuscript. Wrong search strategy may lead to a wrong results and conclusion.

Reviewer #3: A good article attempting to address some factors responsible for the poor response of patients with head and neck cancer with the PNI of patients being portrayed as a potential effective tool to prognosticate these group of patients. However refences 3 and 6 need to be properly cited.

6. PLOS authors have the option to publish the peer review history of their article (what does this mean?). If published, this will include your full peer review and any attached files.

Reviewer #1: No

Reviewer #2: No

Reviewer #3: No

---

## [Author Response · Author response to Decision Letter 0]

26 Jul 2021

Dear editor and dear reviewers,

Thank you for your letter and for the comments concerning our manuscript entitled “Prognostic role of the prognostic nutritional index (PNI) in patients with head and neck neoplasms undergoing radiotherapy: A meta-analysis” (PONE-D-21-13461). Those comments are all valuable and very helpful for revising and improving our article. We have studied comments carefully and have made correction which we hope meet with approval. Revised portions are marked in revision model. The main corrections in the article and responses to the editor’s and reviewers’ comments are as following:

#Editor: 

Comment: “I have biggest concern that search strategy was not appropriate to retrieve all relevant studies.”

Response: It is really a point that should be noticed and the Reviewer 2 also mentioned the similar question. To dispel this concern:

1. the exact electronic search strategy has been attached in Supporting information (S1_File).

Actually, when we conducted literature retrieval systematically, we chose expanded search at the same time, which means it included MeSH terms found below this targeted term in the MeSH hierarchy. That is how we ensure that we can retain all of the relevant articles including all kinds of sub types of head and neck neoplasms.

On top of that, we also traced citations in the reference sections of the retrieved studies so that to make sure no one of eligible studies were left.

2. a detailed explanation has been added in the revised manuscript in section of “2.1 Search strategy and study selection” on page 4.

“We comprehensively searched three main databases, PubMed, Web of Science, and Embase, using a combination of MeSH and entry terms. The exact electronic search strategy has been attached in Supporting information (S1_File). These search strategies were determined using multiple preretrieval tests and would be slightly adjusted when used in different databases. We also conducted expanded search at the same time to ensure that we can retain all of the relevant articles involving all kinds of sub types of head and neck neoplasms. All qualified publications from inception to November 30th, 2020 were searched for. The references of all eligible articles were also independently screened to identify additional studies excluded in the initial search.”

#Reviewer 1:

Comment 1: “Please comment on the PNI threshold, should and how should it clinically be used?”

Response: Yes, it should be wildly used in clinic. Relevant statements have been added in “Discussion” of revised manuscript on page 12.

“PNI threshold is really essential when we use this indicator in clinic to make a prognostication. It is safer for a patient whose PNI is greater than the threshold to receive radiotherapy in statistics. It is a cut-off point currently calculated by three main approaches: ROC curve, Cutoff Finder (a web application), and PNI median. Among them, ROC curve has been gradually recognized as one of the most promising methods, and has been used widely in practice. However, a clear and unified cutoff value of PNI has yet to be decided so far, and it varies in a wide range in different studies (42.7 to 55 in this meta-analysis). That would be a great constraint when we promote this indicator in clinic. Hence, more evidence is urgently required to figure out what is the optimal cutoff value of the PNI.”

Comment 2: “Your conclusion should be rewritten and be formulation more strongly, you’ve looked into all the articles, so you might have a profound opinion about the PNI.”

Response: The rewritten “Conclusion” can be seen on page 14: 

“The PNI is a promising indicator to predict prognostic survival outcomes of HNN patients undergoing radiotherapy. HNN patients with a lower pretreatment PNI are more likely to face a significantly worse OS, DMFS, and PFS, which offers an opportunity to improve the patients’ nutritional and immune statuses before radiotherapy. Future researches are needed to help identify an optimal cutoff value of the PNI, which will guide clinicians to make more precise and individualized therapeutic plans and further improve patients’ survival outcomes.”

Comment 3: “Small comment: please remove the first n=0 in the block diagram and please mention the three databases used. Furthermore, this is a clear and good diagram.”

Response: Proper adjustments have been made in the diagram (Figure 1). 

#Reviewer 2:

Comment: “…there is possibility that, articles may include word head and neck or may not and only use name of specific sub type. How these were managed in the search process is not clear and not clearly indicated in the manuscript.”

Response: It is really a point that should be noticed. To dispel this concern:

1. the exact electronic search strategy has been attached in Supporting information (S1_File).

Actually, when we conducted literature retrieval systematically, we chose expanded search at the same time, which means it included MeSH terms found below this targeted term in the MeSH hierarchy. That is how we ensure that we can retain all of the relevant articles including all kinds of sub types of head and neck neoplasms.

On top of that, we also traced citations in the reference sections of the retrieved studies so that to make sure no one of eligible studies were left.

2. a detailed explanation has been added in section of “2.1 Search strategy and study selection” of the revised manuscript on page 4.

“We comprehensively searched three main databases, PubMed, Web of Science, and Embase, using a combination of MeSH and entry terms. The exact electronic search strategy has been attached in Supporting information (S1_File). These search strategies were determined using multiple preretrieval tests and would be slightly adjusted when used in different databases. We also conducted expanded search at the same time to ensure that we can retain all of the relevant articles involving all kinds of sub types of head and neck neoplasms. All qualified publications from inception to November 30th, 2020 were searched for. The references of all eligible articles were also independently screened to identify additional studies excluded in the initial search.”

#Reviewer 3:

Comment 1: “refences 3 and 6 need to be properly cited.”

Response: Proper adjustments have been done in the revised manuscript.

Comment 2: “To reach an agreement to either accept or reject? Please complete the statement.”

Response: Detailed information has been added in “2.3 Quality assessment” on page 5.

 “The process was conducted by 2 researchers. If any discrepancy appeared, they would discuss with each other, and went back to review the article again. If it didn’t work by discussion to reach an agreement, another researcher would be invited to evaluate the quality of the article independently and finally make a decision.”

Comment 3: “It will be nice to give more elaboration on the radiotherapy schedule and techniques as this will impact differently on patients. For instance, total dose, fractionation, equipment, field number etc all can give different outcomes to the patient.”

Response: Proper adjustments can be seen in “3.1 Retrieval of literature and study characteristics” on page 7.

“All the patients were treated with radiotherapy such as CRT and IMRT.”

We could list the radiotherapy techniques in this way, but more details like total dose are difficult to exhibit here due to great difference in radiotherapy schedules for different sub types of head and neck neoplasms, such as nasopharyngeal carcinoma and cervical esophageal squamous cell carcinoma. Hence, in this meta-analysis, we just pay close attention to whether or not patients received radiotherapy treatment.

Special thanks to all of you for your good comments.

Sincerely yours,

Changgui Kou, Professor, Ph.D.

Department of Epidemiology and Biostatistics,

School of Public Health, Jilin University, 

1163, Xinmin Street, Changchun, 130021 China. 

Tel.: +86 431 85619173; 

fax: +86 431 85645486. 

E-mail: koucg@jlu.edu.cn

---

## [Decision Letter · Decision Letter 1]

1 Sep 2021

Prognostic role of the prognostic nutritional index (PNI) in patients with head and neck neoplasms undergoing radiotherapy: A meta-analysis

PONE-D-21-13461R1

Dear Dr. Kou,

We’re pleased to inform you that your manuscript has been judged scientifically suitable for publication and will be formally accepted for publication once it meets all outstanding technical requirements.

Kind regards,

Mona Pathak, PhD

Academic Editor

PLOS ONE

Additional Editor Comments (optional):

Reviewers' comments:

Reviewer's Responses to Questions

**Comments to the Author**

1. If the authors have adequately addressed your comments raised in a previous round of review and you feel that this manuscript is now acceptable for publication, you may indicate that here to bypass the “Comments to the Author” section, enter your conflict of interest statement in the “Confidential to Editor” section, and submit your "Accept" recommendation.

Reviewer #2: All comments have been addressed

Reviewer #3: All comments have been addressed

2. Is the manuscript technically sound, and do the data support the conclusions?

Reviewer #2: Yes

Reviewer #3: Yes

3. Has the statistical analysis been performed appropriately and rigorously? 

Reviewer #2: Yes

Reviewer #3: Yes

4. Have the authors made all data underlying the findings in their manuscript fully available?

Reviewer #2: Yes

Reviewer #3: Yes

5. Is the manuscript presented in an intelligible fashion and written in standard English?

Reviewer #2: Yes

Reviewer #3: Yes

6. Review Comments to the Author

Reviewer #2: (No Response)

Reviewer #3: Well written article . Previously raised issues had been addressed except that reference number 2 was incomplete. I recommend that the reference be revisited and rewritten as appropriate.

7. PLOS authors have the option to publish the peer review history of their article (what does this mean?). If published, this will include your full peer review and any attached files.

Reviewer #2: No

Reviewer #3: **Yes: **DR ADMU ABDULLAHI

---

## [Editor Report · Acceptance letter]

6 Sep 2021

PONE-D-21-13461R1 

Prognostic role of the prognostic nutritional index (PNI) in patients with head and neck neoplasms undergoing radiotherapy: A meta-analysis 

Dear Dr. Kou:

I'm pleased to inform you that your manuscript has been deemed suitable for publication in PLOS ONE. Congratulations! Your manuscript is now with our production department. 

Kind regards, 

on behalf of

Dr. Mona Pathak 

Academic Editor

PLOS ONE